# Security Verification of an Authentication Algorithm Based on Verifiable Encryption

Maki Kihara *,† and Satoshi Iriyama †

Department of Information Science, Tokyo University of Science, Yamazaki 2641, Noda 278-8510, Chiba, Japan
* Correspondence: kiharamaki18@gmail.com
† These authors contributed equally to this work.

**Abstract:** A new class of cryptosystems called verifiable encryption (VE) that facilitates the verification of two plaintexts without decryption was proposed in our previous paper. The main contributions of our previous study include the following. (1) Certain cryptosystems such as the one-time pad belong to the VE class. (2) We constructed an authentication algorithm for unlocking local devices via a network that utilizes the property of VE. (3) As a result of implementing the VE-based authentication algorithm using the one-time pad, the encryption, verification, and decryption processing times are less than 1 ms even with a text length of 8192 bits. All the personal information used in the algorithm is protected by Shanon's perfect secrecy. (4) The robustness of the algorithm against man-in-the-middle attacks and plaintext attacks was discussed. However, the discussion about the security of the algorithm was insufficient from the following two perspectives: (A) its robustness against other theoretical attacks such as ciphertext-only, known-plaintext, chosen-plaintext, adaptive chosen-plaintext, chosen-ciphertext, and adaptive chosen-ciphertext attacks was not discussed; (B) a formal security analysis using security verification tools was not performed. In this paper, we analyze the security of the VE-based authentication algorithm by discussing its robustness against the above theoretical attacks and by validating the algorithm using a security verification tool. These security analyses, show that known attacks are ineffective against the algorithm.

**Keywords:** verifiable encryption; authentication; one-time pad; cryptography; information security

## 1. Introduction

With a significant increase in the number of Internet users, network technologies have advanced significantly, including 5th-generation (5G) wireless, virtual reality (e.g., the metaverse), and the Internet of Things (IoT). Moreover, the concept of smart cities is spreading around the world, and various network technologies have been used widely to solve many problems. In particular, IoT devices have been adopted in various domains [1–3] and the development of IoT technology is said to be one of the best ways to construct a smart community [4].

In light of this scenario, there is an increase in the number of authentications required for using such Internet services. However, the number of passwords is far less than the number of Internet services because users tend to reuse the same passwords. The leakage of reused passwords can be exploited by multiple services that use the same password. Additionally, users often set simple and easy-to-remember passwords, including personal information, such as phone numbers, addresses, and birthdays; celebrity names; or nouns in the dictionary. In such cases, there is the potential for unauthorized logins through password guessing and dictionary attacks. In fact, unauthorized logins to social network services, online shopping sites, and IoT devices occur frequently. To solve these issues, many authentication mechanisms have been studied, including authentication protocols using cryptographic techniques such as secret sharing and secret computation, systems combining cryptography and steganography, protocols using Captcha to prove that the

user being authenticated is not a computer, biometric authentication, and multi-factor authentication [5–9]. Furthermore, authentication in virtual reality and authentication systems for 5G have been studied extensively [10,11].

Similarly, many authentication schemes for IoT have been studied, including biometric authentication and authentication based on blockchain technology [12–15]. According to [14], IoT security issues can be attributed to the limited computational resources of devices, which make it difficult to apply general cryptosystems that require high computational power for security, such as those used in traditional applications, and may reduce the convenience of many devices.

In other words, lightweight and secure authentication algorithms are required for IoT devices with limited computational resources .

For these reasons, in 2019 [16] we proposed an authentication algorithm for IoT devices. We assumed that the authentication of IoT devices was used for unlocking devices accessed by multiple people, rather than for devices accessed by only one person. The proposed algorithm was based on the properties of a special class of cryptosystems called verifiable encryption (VE), which facilitates the calculation of a metric between two plain texts without decryption. It has been demonstrated in [16,17] that certain cryptosystems such as one-time pads and the identification algorithm proposed by Schnorr [18] belong to the class of VE. In implementing our proposed authentication algorithm with the one-time pad, we achieved an authentication speed of 0.1 ms or less, even with a text length of 8192 bits, and it was robust against cleartext attacks [16].

However, the security of this algorithm is yet to be investigated in terms of robustness to other theoretical attacks, and it has not been analyzed using formal security protocol verification tools such as ProVerif [19]. In other words, our previous paper [16] did not sufficiently discuss the security of the algorithm.

The objective of this study was to investigate the security of the proposed authentication algorithm based on VE more comprehensively. To this end, we discuss the robustness of the algorithm against well-known theoretical attacks and conduct security verification using ProVerif, which is a well-known automated verification tool for security properties. The remainder of this paper is organized as follows.

Section 2: The cryptosystem and VE are defined as mathematical preparations.
Section 3: The VE-based authentication algorithm proposed in [16] is introduced.
Section 4: The security of VE-based algorithm is discussed.
Section 5: The findings are summarized.

## 2. Preliminaries

In this section, we define cryptosystems and VE as mathematical preliminaries.

### 2.1. Cryptosystem

Let $\mathcal{P}, \mathcal{C}$, and $\mathcal{K}$ be spaces of plaintexts, ciphertexts, and keys, respectively. A set of encryptions $\mathcal{E}$ and set of decryptions $\mathcal{D}$ are defined as

$$\mathcal{E} = \{E_k : \mathcal{P} \to \mathcal{C} | k \in \mathcal{K}\}, \quad \mathcal{D} = \{D_k : \mathcal{C} \to \mathcal{P} | k \in \mathcal{K}\}.$$

**Definition 1** (Cryptosystem). *A 5-tuple* $(\mathcal{P}, \mathcal{C}, \mathcal{K}, \mathcal{E}, \mathcal{D})$ *is called a cryptosystem if for all* $p \in \mathcal{P}$, *there is a* $k \in \mathcal{K}$ *such that*

$$D_k(E_k(p)) = p.$$

### 2.2. Verifiable Encryption

Let $V : \mathcal{P} \times \mathcal{P} \to \mathbb{R}_+(= [0, +\infty))$ be a metric between two texts.

**Definition 2** (VE). *Given a plaintext space $\mathcal{P}$, metric $V$, and encryption function $E$, we call an encryption function $E$ a VE if for both the plaintexts $p_1, p_2 \in \mathcal{P}$ and keys $k_1, k_2 \in \mathcal{K}$, there are two maps $F : \mathcal{C} \times \mathcal{C} \to \mathcal{C}$ and $D : \mathcal{C} \to \mathbb{R}_+$ such that*

$$D_{k_1,k_2}(F(E_{k_1}(p_1), E_{k_2}(p_2))) = V(p_1, p_2).$$

The two maps $F$ and $D$ depend on on the cryptosystem used. Because $F$ is a map from a cartesian product of two ciphertext spaces $\mathcal{C} \times \mathcal{C}$ to the ciphertext space $\mathcal{C}$ and does not require any keys for any operations, it can realize secure computation to derive the difference between plaintexts.

As mentioned in the introduction, VE facilitates the computation of differences between two plaintexts $p_1$ and $p_2$ without decryption by applying a special function to two ciphertexts $E_{k_1}(p_1)$ and $E_{k_2}(p_2)$ using a composite map $D \circ F$. Therefore, VE has properties similar to homomorphic encryption (HE), Rivest–Shamir–Adleman (RSA) encryption [20], Paillier encryption [21], and fully HE [22], which facilitates computations in the ciphertext space without decryption. HE and VE have the same properties in terms of using homomorphism but produce different outputs following decryption. The output of HE is the result of an operation (addition or multiplication) on two plaintexts, whereas VE derives the distance between two plaintexts.

As proof of example about cryptosystems belonging to VE, we introduce the following theorem proved in paper [16].

**Theorem 1.** *The one-time pad cryptosystem belongs to the class of VE.*

**Proof of Theorem 1.** The one-time pad cryptosystem can be described as follows:

$$C = (\mathcal{P}, \mathcal{C}, \mathcal{K}, \mathcal{E}, \mathcal{D}) \quad \text{where}$$
$$\mathcal{P} = \mathcal{C} = \mathcal{K} = \{0,1\}^n,$$
$$\mathcal{E} = \{E_k | E_k(p) = p_i + k_i \mod 2 (\forall i \in \{1, \cdots, n\}) = p \oplus k, p \in \mathcal{P}, k \in \mathcal{K}\},$$
$$\mathcal{D} = \{D_k | D_k(c) = c_i + k_i \mod 2 (\forall i \in \{1, \cdots, n\}) = c \oplus k, c \in \mathcal{C}, k \in \mathcal{K}\},$$

where an arbitrary key $k \in \mathcal{K}$ is randomly chosen from $\{0,1\}^n$ according to a uniform distribution. Here, $\oplus$ denotes a bitwise XOR operation.

Let $p_1, p_2 \in \mathcal{P}$ be plaintexts. We define the Hamming distance $V(p_1, p_2)$ as follows:

$$V(p_1, p_2) = |\{i \in \{1, 2, \cdots, n\} | p_{1,i} \neq p_{2,i}, \quad p_1, p_2 \in \mathcal{P}\}|$$

The Hamming distance is a count of the number of bit differences. If $p_{1,i} = p_{2,i}$, then $p_{1,i} \oplus p_{2,i} = 0$. Otherwise (i.e., $p_{1,i} \neq p_{2,i}$), $p_{1,i} \oplus p_{2,i} = 1$.

If $s = p_1 \oplus p_2 = (p_{1,1} \oplus p_{2,1})(p_{1,2} \oplus p_{2,2}) \cdots (p_{1,n} \oplus p_{2,n}) = s_1 s_2 \cdots s_n$, then the $V(p_1, p_2)$ defined above can be rewritten as

$$
\begin{aligned}
V(p_1, p_2) &= |\{i \in \{1, 2, \cdots, n\} | p_{1,i} \neq p_{2,i}, \quad p_1, p_2 \in \mathcal{P}\}| \\
&= (p_{1,1} \oplus p_{2,1}) + (p_{1,2} \oplus p_{2,2}) + \cdots + (p_{1,n} \oplus p_{2,n}) \\
&= s_1 + s_2 + \cdots + s_n \\
&= \sum_{i=1}^{n} s_i \\
&= \sum_{i=1}^{n} (p_1 \oplus p_2)_i.
\end{aligned}
$$

The definition of VE is satisfied if $V$ is the correct Hamming distance. Let $p_1, p_2 \in \mathcal{P}$ be two plaintexts, $k_1, k_2 \in \mathcal{K}$ be two keys, and $c_1 = E_{k_1}(p_1) = p_1 \oplus k_1, c_2 = E_{k_2}(p_2) =$

$p_2 \oplus k_2 \in \mathcal{C}$ be two ciphertexts. Then, we can construct $F : \mathcal{C} \times \mathcal{C} \to \mathcal{C}$ and $D : \mathcal{C} \to \mathbb{R}_+$ as follows:

$$F(c_1, c_2) := c_1 \oplus c_2 = E_{k_1}(p_1) \oplus E_{k_2}(p_2) = (p_1 \oplus k_1) \oplus (p_2 \oplus k_2)$$

$$D_{k_1, k_2}(c) := \sum_{i=1}^{n} (c \oplus k_1 \oplus k_2)_i.$$

We then calculate

$$D_{k_1, k_2}(F(E_{k_1}(p_1), E_{k_2}(p_2))) = \sum_{i=1}^{n} (F(c_1, c_2) \oplus k_1 \oplus k_2)$$

$$= \sum_{i=1}^{n} ((c_1 \oplus c_2) \oplus k_1 \oplus k_2)_i$$

$$= \sum_{i=1}^{n} (((p_1 \oplus k_1) \oplus (p_2 \oplus k_2)) \oplus k_1 \oplus k_2)_i$$

$$= \sum_{i=1}^{n} ((p_1 \oplus p_2) \oplus (k_1 \oplus k_1) \oplus (k_2 \oplus k_2))_i$$

$$= \sum_{i=1}^{n} (p_1 \oplus p_2)_i = V(p_1, p_2).$$

Therefore, the one-time pad cryptosystem belongs to the VE class. □

In the case of the one-time pad cryptosystem, $F, D$ and $V$ have the configurations described above. However, the configuration of $F, D$ and $V$ varies depending on the cryptosystem used.

### 3. Authentication Algorithm Based on VE

In this section, we introduce the authentication algorithm based on the VE proposed in [16]. First, we describe the general structure of authentication. Let Alice be a user to be authenticated and Bob be an authenticator. We consider that the authentication between Alice and Bob includes the following two steps:

**Registration** Alice distributes her personal information to Bob.

**Verification** Bob checks if the personal information distributed by Alice in advance matches the personal information sent from Alice for verification.

The assumptions adopted in our previously proposed algorithm are as follows:

- Alice is the user to be authenticated.
- Bob is an authenticator and trusted by Alice.
- S is a server and untrusted by Alice.
- The channel between Alice and Bob is a secure channel that includes direct access.
- The channel between Bob and S is an insecure channel.
- Alice's personal information is never provided to S.

We now describe the authentication algorithm based on the VE proposed in [16]. Note that the algorithm is generalized because the maps $F, D$ and $V$ depend on the cryptosystem used.

Let $C = (\mathcal{P}, \mathcal{C}, \mathcal{K}, \mathcal{E}, \mathcal{D})$ be a cryptosystem and $E \in \mathcal{E}$ be a VE. Let $p_1, p_2 \in \mathcal{P}$ be two plain texts, $k_1, k_2 \in \mathcal{K}$ be two keys, and

$$c_1 = E_{k_1}(p_1) \in \mathcal{C},$$
$$c_2 = E_{k_2}(p_2) \in \mathcal{C}$$

be two ciphertexts.

**[Registration step (Figure 1)]**

1. Alice sends her personal information $p_1$ to Bob.
2. Bob generates a key $k$ and calculates $c_1 = E_{k_1}(p_1)$.
3. Bob sends $c_1$ to server S and discards $p_1$.

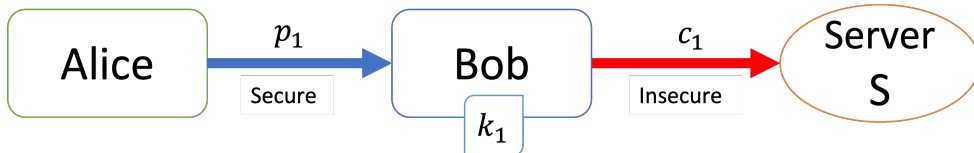

**Figure 1.** Registration step.

**[Verification step (Figure 2)]**

1. Alice sends her personal information $p_2$ to Bob.
2. Bob generates a key $k_2$ and calculates $c_2 = E_{k_2}(p_2)$.
3. Bob sends $c_2$ to S.
4. Server S calculates $c_d = F(c_1, c_2)$ and sends $c_d$ to Bob.
5. Bob calculates the result $r = D_{k_1,k_2}(c_d)$ and checks $r$.

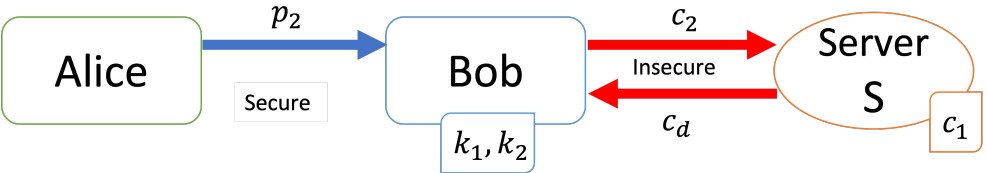

**Figure 2.** Verification step.

The VE-based authentication algorithm for unlocking local devices through a network requires neither a certificate authority (CA) nor a public key agreement (PKA).

Next, we present an example of applying the one-time pad, which is one of the cryptosystems belonging to VE, to the proposed algorithm. The results of this implementation are presented in [16].

Let Alice be a user to be authenticated, Bob be an authenticator and trusted by Alice, and the server be an untrusted party. If Bob is an IoT device, assume that the channel between Alice and Bob is implicitly considered a secure channel (e.g., direct access). If Alice and Bob cannot physically contact each other, then a secure channel must be prepared.

Let $C = (\mathcal{P}, \mathcal{C}, \mathcal{K}, \mathcal{E}, \mathcal{D})$ be a one-time pad cryptosystem defined as

$$\mathcal{P} = \mathcal{C} = \mathcal{K} = \{0,1\}^n,$$
$$\mathcal{E} = \{E_k | E_k(p) = p \oplus k, p \in \mathcal{P}, k \in \mathcal{K}\},$$
$$\mathcal{D} = \{D_k | D_k(c) = c \oplus k, c \in \mathcal{C}, k \in \mathcal{K}\}.$$

Let $p_1, p_2 \in \mathcal{P}$ be two plaintexts, $k_1, k_2 \in \mathcal{K}$ be two keys, and $c_1, c_2$ be two cipher texts. Additionally, $V, F$ and $D$ are defined as follows:

$$V(p_1, p_2) = \sum_{i=1}^{n}(p_1 \oplus p_2)_i,$$
$$F(c_1, c_2) := c_1 \oplus c_2 = (p_1 \oplus k_1) \oplus (p_2 \oplus k_2),$$
$$D_{k_1,k_2}(c) := \sum_{i=1}^{n}(c \oplus k_1 \oplus k_2)_i.$$

**[Registration step (Figure 3)]**

1. Alice sends her personal information $p_1$ to Bob.
2. Bob generates a key $k_1$ and computes $c_1 = p_1 \oplus k_1$.
3. Bob sends $c_1$ to server S and discards $p_1$, and S stores $c_1$ in its database.

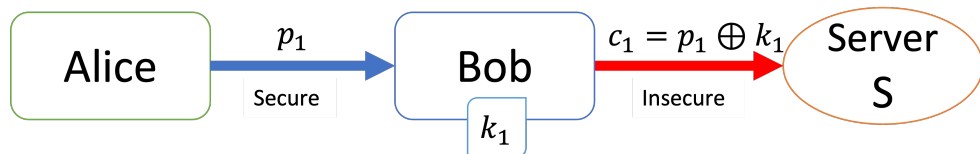

**Figure 3.** Registration step with the one-time pad.

**[Verification step (Figure 4)]**

1. Alice sends her personal information $p_2$ to Bob.
2. Bob generates a one-time key $k_2$ and computes $c_2 = p_2 \oplus k_2$.
3. Bob sends $c_2$ to server S.
4. S computes the encrypted result $c_d = F(c_1, c_2)$, where

$$c_d = F(c_1, c_2)$$
$$= c_1 \oplus c_2$$
$$= (p_1 \oplus k_1) \oplus (p_2 \oplus k_2),$$

and sends $c_d$ to Bob.

5. Bob computes the result $r = D_{k_1,k_2}(F(c_1, c_2))$ to obtain the distance $V(p_1, p_2)$ as

$$r = D_{k_1,k_2}(F(c_1, c_2)) = \sum_{i=1}^{n}(F(c_1, c_2) \oplus k_1 \oplus k_2)_i$$
$$= \sum_{i=1}^{n}((c_1 \oplus c_2) \oplus k_1 \oplus k_2)_i$$
$$= \sum_{i=1}^{n}(((p_1 \oplus k_1) \oplus (p_2 \oplus k_2)) \oplus k_1 \oplus k_2)_i$$
$$= \sum_{i=1}^{n}((p_1 \oplus p_2) \oplus (k_1 \oplus k_1) \oplus (k_2 \oplus k_2))_i$$
$$= \sum_{i=1}^{n}(p_1 \oplus p_2)_i = V(p_1, p_2)$$

and checks the result $r$. If $r = 0$, then Bob returns "OK" to Alice. Otherwise Bob returns "NG".

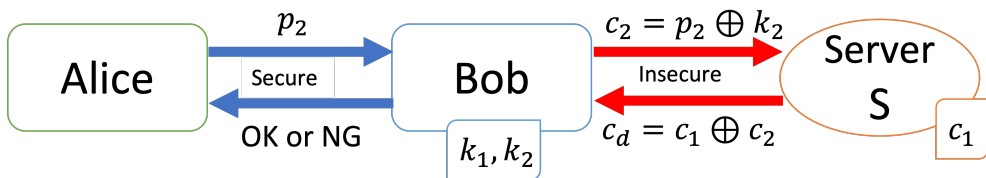

**Figure 4.** Verification step with the one-time pad.

This example does not require CA or PKA, similar to the algorithm based on VE. Furthermore, because most of the calculations are XOR operations, the computational burden is very small. The execution speed of this authentication mainly depends on the time required for key generation. In our previous study [16], the processing time required for encryption, verification, and decryption was less than 1 ms, even with a text length 8192 bits. Please refer to [16] for details regarding the experimental results.

## 4. Robustness of the Authentication Algorithm Based on VE

We previously reported that the VE-based authentication algorithm is robust to plaintext attacks and man-in-the-middle attacks [16]. However, its robustness to other theoretical attacks has not been investigated. Additionally, the security of the proposed algorithm has not been verified using security protocol verification tools.

In this section, we discuss the robustness of the VE-based authentication algorithm by applying the one-time pad against some theoretical attacks and verify the security of the algorithm using the automatic security verification tool ProVerif [19]. Here, the objective of the attacker, Eve, is to obtain Alice's personal information. Additionally, we compare our algorithm to other authentication technologies for the IoT environments introduced in [14].

*4.1. Theoretical Attacks*

Here, we discuss the robustness of the proposed VE-based authentication algorithm with the one-time pad against several theoretical attacks, namely ciphertext-only, known-plaintext, chosen-plaintext, adaptive chosen-plaintext, chosen-ciphertext, and adaptive chosen-ciphertext attacks. Note that we discuss implementations using a one-time pad cryptosystem, rather than a generalized algorithm. We referred to [23] for the definition of each attack.

4.1.1. Ciphertext-Only Attack

We assume that Eve knows only one ciphertext. It should be noted that Eve can attempt a brute force attack against our algorithm to perform decryption using all the keys in the key space. However, because our implementation uses the one-time pad, the ciphertext-only attack by Eve will not succeed because of Shannon's perfect secrecy. In addition, by increasing $n$, which is the dimension of the key space, we can ensure that the probability that different plaintexts can be encrypted using the same key is very small. Therefore, the attack is not successful.

Next, we consider the case where Eve obtains all ciphertexts. Let $p_1, p_2 \in \mathcal{P}$ be two plain texts, $k_1, k_2 \in \mathcal{K}$ be two keys, and

$$c_1 = E_{k_1}(p_1) = p_1 \oplus k_1 \in \mathcal{C},$$
$$c_2 = E_{k_2}(p_2) = p_2 \oplus k_2 \in \mathcal{C}$$

be two cipher texts. Additionally, we provide two maps $F : \mathcal{C} \times \mathcal{C} \to \mathcal{C}$ and $D : \mathcal{C} \to \mathbb{R}_+$ as follows:

$$F(c_1, c_2) := c_1 \oplus c_2,$$
$$D_{k_1,k_2}(c) := \sum_{i=1}^{n} (c \oplus k_1 \oplus k_2)_i.$$

Therefore, we assume that Eve can obtain

$$c_1 = E_{k_1}(p_1) = p_1 \oplus k_1$$
$$c_2 = E_{k_2}(p_2) = p_2 \oplus k_2$$
$$c_d = F(c_1, c_2) = c_1 \oplus c_2 = (p_1 \oplus k_1) \oplus (p_2 \oplus k_2).$$

The information that Eve can obtain by combining ciphertexts $c_1, c_2$, and $c_d$ are as follows.

According to the results presented in Table 1, even if Eve obtains all ciphertexts, it is not possible to obtain information regarding Alice's plaintexts $p_1$ or $p_2$.

**Table 1.** Information that Eve can obtain by combining ciphertexts.

| Combination of Ciphertexts | Calculation by Eve | Result |
|:---:|:---:|:---:|
| $c_1$ and $c_2$ | $c_1 \oplus c_2$ | $c_d$ |
| $c_1$ and $c_d$ | $c_1 \oplus c_d$ | $c_2$ |
| $c_2$ and $c_d$ | $c_2 \oplus c_d$ | $c_1$ |
| All ciphertexts | $c_1 \oplus c_2 \oplus c_d$ | 0 |

However, according to the security verification presented in the previous paper [16], we know that if key $k_2$ is equal to $k_1$ or if $k_2$ is not a one-time-key (i.e., the same key is reused for every verification), then the security of the algorithm is not guaranteed. In the former case ($k_2$ is equal to $k_1$), if Eve acquires $c_1$ at the time of registration and $c_2$ at the time of verification, then Eve can obtain the metric for $p_1$ and $p_2$ through the following calculation:

$$
\begin{aligned}
c_1 \oplus c_2 &= (p_1 \oplus k_1) \oplus (p_2 \oplus k_2) \\
&= (p_1 \oplus k_1) \oplus (p_2 \oplus k_1) \quad (\because k_2 = k_1) \\
&= (p_1 \oplus p_2) \oplus (k_1 \oplus k_1) \\
&= p_1 \oplus p_2.
\end{aligned}
$$

Furthermore, suppose that Eve acquires $c_1$ in the registration step and then sends $p_2 = 0$ in the verification step to impersonate Alice. In this case, because $c_2$ is equal to $k_1$, Eve can obtain $c_1 \oplus k_1 = p_1$ easily.

In the latter case ($k_2$ is reused for every verification), if Eve impersonates Alice and sends $p_2 = 0$ to Bob, then $c_2$ is equal to $k_2$. Therefore, Eve can eavesdrop on $k_2$ from an insecure channel (i.e., the channel between Bob and the server). If Eve obtains $c_2$ sent by Bob in a subsequent verification, then Eve can easily obtain Alice's secret information $p_2$ by using the previously obtained $k_2$.

Therefore, to maintain the security of the algorithm, it is essential to ensure that $k_1$ and $k_2$ are different keys and that $k_2$ is a one-time key.

### 4.1.2. Known-Plaintext Attack

We assume that Eve knows the plaintexts and ciphertexts that correspond to the plaintexts, excluding Alice's plaintext. Eve attempts to obtain Alice's secret information from some pairs of plaintexts and ciphertexts that Eve already knows.

This is a known-plaintext attack against the VE-based authentication algorithm. If Eve knows a pair of plaintext and ciphertext instances, then Eve can easily find the corresponding key because the one-time pad cryptosystem is a symmetric key cryptosystem utilizing XOR operations. Let $p \in \mathcal{P}$ be a plaintext, $k \in \mathcal{K}$ be a key, and $c = E_k(p) = p \oplus k \in \mathcal{C}$ be a ciphertext. The corresponding key is obtained through the following calculation:

$$
\begin{aligned}
c \oplus p &= (p \oplus k) \oplus p \\
&= (p \oplus p) \oplus k \\
&= k.
\end{aligned}
$$

Therefore, it can be said that Eve knows some triplets of plaintexts, ciphertexts, and keys.

However, because we apply the one-time pad to the algorithm, the key obtained by Eve and the key used to encrypt Alice's plaintext are different. Therefore, regardless of the extent to which Eve knows ciphertext-plaintext pairs, it is difficult to obtain Alice's plaintext from the information obtained by Eve if Bob does not reuse the same key.

### 4.1.3. Chosen-Plaintext Attack and Adaptive Chosen-Plaintext Attack

We assume that Eve can encrypt a plaintext of her choice before knowing which ciphertext she wishes to decrypt. In other words, Eve can obtain a ciphertext for an arbitrary plaintext other than Alice's secret information. In the case of an arbitrary public key cryptosystem, it can be assumed that Eve does not know the decryption key (i.e., secret key). However, Eve knows the decryption key because the one-time pad is a symmetric key cryptosystem. In other words, Eve knows some triplets of plaintexts, ciphertexts, and the keys. Eve attempts to obtain Alice's secret from some triplets that Eve already knows.

In the case of a chosen-plaintext attack against the VE-based algorithm, even if Eve can encrypt all the plaintexts other than Alice's plaintext, Eve cannot obtain Alice's plaintext because the keys obtained by Eve and the key used to encrypt Alice's plaintext are different.

Therefore, it can be concluded that it is difficult for Eve to obtain Alice's plaintext, similar to the known-plaintext attack.

Furthermore, in an adaptive chosen-plaintext attack, Eve can encrypt the plaintext of her own choice after knowing which ciphertext she wishes to decrypt. The difference between the chosen-plaintext attack and the adaptive chosen-plaintext attack is whether Eve knows the ciphertext that corresponds to Alice's secret information prior to encrypting an arbitrary plaintext other than Alice's plaintext. Therefore, the result is the same as that of the chosen-plaintext attack.

### 4.1.4. Chosen-Ciphertext Attack and Adaptive Chosen-Ciphertext Attack

We assume that an attacker Eve can decrypt the ciphertexts of her choice, even if she does not know the relevant key. In other words, Eve can obtain plaintext for an arbitrary ciphertext other than Alice's plaintext and the ciphertext that corresponds to Alice's secret information without keys.

In the case of a one-time pad, which is a symmetric key cryptosystem, this attack is similar to the known-plaintext and chosen-plaintext attacks. Eve can only acquire triplets of plaintexts, ciphertexts, and keys. Therefore, it is difficult for Eve to obtain Alice's plaintext, similar to the known-plaintext and chosen-plaintext attacks.

Furthermore, the difference between the chosen-ciphertext attack and adaptive chosen-ciphertext attack is similar to a chosen-plaintext attack. Specifically, Eve must acquire the ciphertext that corresponds to Alice's secret information before decrypting an arbitrary ciphertext other than Alice's plaintext. Therefore, both chosen-ciphertext attacks and adaptive chosen-ciphertext attacks are ineffective against the algorithm based on VE.

### 4.2. Security Verification Using ProVerif

Here, we model and analyze the authentication algorithm based on VE using the automatic security verification tool ProVerif [19] for the cryptographic protocols developed by Bruno Blanchet. Because the proposed algorithm can be applied not only to a specific cryptographic system but also to other cryptographic systems belonging to the VE class, ProVerif, which enables symbolic verification, was adopted.

An explanation of each variable is given below.

**Variables**

```
free p1: Plaintext[private]. (*plaintext of Alice for registration*)
free p2: Plaintext[private]. (*plaintext of Alice for verification*)
free k1: key[private]. (*key to encrypt p1*)
Free AB: channel[private] (*between Alice and Bob*)
free BS: channel. (*between Bob and server*)
```

Here, p1 and p2 correspond to $p_1$ and $p_2$ in the algorithm described in Section 3, respectively. Additionally, k1 corresponds to $k_1$ in the algorithm. However, $k_2$ is not introduced here as a variable because $k_2$ is not sent anywhere after it is generated by Bob and is discarded after one verification. Additionally, AB represents the channel between Alice and Bob, and BS represents the channel between Bob and the server.

### 4.2.1. Verification Summary of the Registration Process

Here, we present a verification summary of the registration process of the algorithm.

```
RESULT not attacker(p1[]) is true.
RESULT not attacker(k1[]) is true.
RESULT Weak secret p1 is true.
RESULT event(BgetP1(id,p_1))
==> event(AsendP1(id,p_1)) is true.
RESULT inj-event(SgetC1(id,c))
==> inj-event(BsendC1(id,c)) is false.
RESULT (even event(SgetC1(id,c))
==> event(BsendC1(id,c)) is false.)
```

The first and second lines indicate that the attacker cannot obtain Alice's plaintext $p_1$ and key $k_1$ in the registration phase. In the third line, we check the possibility of offline attacks against $p_1$. Even if the entropy of secret information $p_1$ is low, meaning $p_1$ is a string that humans can interpret, the attacker still cannot reach $p_1$. In the fourth line, we check if Alice is sending $p_1$ (corresponds to the event "AsendP1") before Bob obtains $p_1$ (corresponds to "BgetP1"), and the result is "true." Therefore, it is clear that the operation is completed correctly. In the fifth line, we check if Bob is sending $c_1$ (corresponds to "BsendC1") before the server obtains $c_1$ (corresponds to "SgetC1") to determine the possibility of an attack, and the result is "false". This query verifies whether the ciphertext $c_1$ received by the server is sent by Bob. An attacker can impersonate Bob or the server because the channel between them is not secure. Therefore, this query is "false".

Here, we consider some possible attacks in the registration phase. In this case, Eve is located between Bob and the server, but Eve can only obtain the ciphertext $c_1$ that encrypts a plaintext $p_1$ using the key $k_1$, and neither $p_1$ nor $k_1$ can be obtained.

First, Eve attempts to obtain Alice's plaintext $p_1$ from $c_1$. It is difficult for Eve to obtain $p_1$ because only $c_1$ is given to Eve and $k_1$ is never sent anywhere. However, if Bob repeatedly uses $k_1$ during the registration phase for a user who is not Alice, then cryptoanalysis is possible using a statistical method.

Second, Eve impersonates Bob and sends $c_1'$ instead of $c_1$ to the server. This attack can be considered a man-in-the-middle attack. In this case, the authentication of Alice will always fail, even if Alice sends the correct plaintext to Bob during the verification phase. Furthermore, the same result will occur when Eve impersonates a server or when she impersonates Bob. In this case, Alice becomes a user who is not registered at the time of verification. Therefore, even if the correct plaintext is sent, Alice's authentication will fail. This attack only causes the failure of Alice's authentication and Eve cannot obtain Alice's personal information.

Therefore, such attacks are ineffective.

### 4.2.2. Summary of the Verification Process

Here, we present a summary of the verification process of the algorithm.

```
RESULT not attacker(p2[]) is true.
RESULT not attacker(k1[]) is true.
RESULT Weak secret p2 is true.
RESULT event(BgetP2(id_1,p_1))
==>event(AsendP2(id_1,p_1)) is true.
RESULT inj-event(SgetC2(id_1,c_1))
==>inj-event(BsendC2(id_1,c_1)) is false.
RESULT (even event(SgetC2(id_1,c_1))
==>event(BsendC2(id_1,c_1)) is false.)
RESULT inj-event(BgetCD(c_1))
==> inj-event(SsendCD(c_1)) is false.
RESULT (even event(BgetCD(c_1))
==> event(SsendCD(c_1)) is false.)
RESULT event(AgetON(r_1))
==> event(BsendON(r_1)) is true.
```

The first and second lines demonstrate that the attacker cannot obtain Alice's plaintext $p_2$, which is sent for verification, or key $k_1$, which is used to encrypt $c_1$ in the verification phase. In the third line, we check the possibility of offline attacks against $p_2$. The result is that the attacker cannot reach $p_2$. In the fourth line, we check if Alice is sending $p_2$ (corresponds to the event "AsendP2") before Bob obtains $p_2$ (corresponds to the event "BgetP2"), and the result is "true". In the fifth line, we check if Bob is sending $c_2$ (corresponds to the event "BsendC2") before the server obtains $c_2$ (corresponds to the event "SgetC2") to determine whether an attack is possible, and the result is "false". Similar to registration, an attacker can impersonate Bob or the server because the channel between them is not secure. Therefore, this query is "false". In the seventh line, we check if the server is sending

$c_d$ (corresponds to the event "SsendCD") before it obtains $c_d$ (corresponds to the event "BgetCD") to determine the possibility of an attack, and the result is "false". This result is consistent with that of the fifth line.

Here, we consider some possible attacks during the verification process. Similar to the registration process, we assume that Eve is located between Bob and the server and that Eve can only obtain the ciphertext $c_2$ that encrypts a plaintext $p_2$ using a key $k_2$ and encrypted metric $c_d$, and that not all plaintexts $p_1, p_2$ or all keys $k_1, k_2$ can be obtained.

First, we assume that Eve attempts to obtain Alice's plaintext $p_2$ from $c_2$. Similar to the registration process, it is difficult for Eve to obtain $p_2$. However, if Bob uses $k_2$, which is equal to $k_2$ during the verification phase, then Eve can obtain the metric for $p_1$ and $p_2$, as mentioned in Section 4.1.1. Furthermore, $k_2$ is reused for every verification and Eve can also obtain $p_1$, as mentioned in Section 4.1.1.

Second, we assume that Eve impersonates Bob and sends $c_2'$ instead of $c_2$ to the server. In this case, the authentication of Alice will always fail. Third, we consider a case where Eve impersonates the server and sends $c_d'$ instead of $c_d$ to Bob. Even if Eve obtains $c_1$ and $c_2$ prior to obtaining $c_d$, Eve still cannot obtain Alice's secret information $p_1$ and $p_2$, as mentioned in Section 4.1.1. If Eve does not obtain $c_1$ and $c_2$, then Eve can prevent Alice's authentication from succeeding by sending $c_d'$ instead of $c_d$ to Bob. However, Eve cannot steal Alice's secret information $p_1$ and $p_2$.

Therefore, these attacks are ineffective.

### 4.3. Comparison

Ref. [14] provides a detailed comparison of the proposed authentication schemes. Here, we categorize the authentication schemes presented in [14] into groups based on the cryptosystems used and we compare them to our algorithm. Our algorithm is generalized and does not specify a cryptosystem. Therefore, we consider the implementation with the one-time pad for security comparisons.

In [14], among the 84 proposed authentication schemes, 44 schemes are based on some public key cryptosystem (e.g., Diffie-Hellman, RSA), 26 schemes are symmetric key cryptosystems (e.g., AES), and 14 schemes are other types (e.g., Hash function).

The security of general cryptosystems, including both symmetric key cryptosystems and public key cryptosystems, depends on computational complexity, whereas the security of the one-time pad depends on perfect secrecy. Therefore, our algorithm is more secure in terms of the security of the cryptosystem used. Additionally, because our algorithm does not require key distribution, there is no need to consider the distribution problem of the one-time pad.

The advantages of our algorithm other than security can be summarized as follows.

1. Our algorithm does not require significant computational resources on devices because all calculations are simple XOR and addition operations.
2. Many authentication schemes introduced in [14] specify the form of plaintext. By contrast, arbitrary plain texts (e.g., biometric information, images, radio-frequency identification, and credit cards) can be processed by our algorithm if personal information can be converted into binary sequences because our algorithm utilizes a generalized form.

One disadvantage of the one-time pad is that the plaintext length and key length must be the same. Therefore, the algorithm is not suitable for low-resource devices when processing very large plaintexts.

## 5. Conclusions

In this paper, we evaluated the security of our previously proposed authentication algorithm based on VE. The proposed algorithm with the one-time pad is robust to theoretical attacks, including ciphertext-only, known-plaintext, chosen-plaintext, adaptive chosen-plaintext, chosen-ciphertext, and adaptive chosen-ciphertext attacks. These theoretical attacks are generally ineffective against our algorithm. However, it is not possible to

maintain security if encryption keys are reused instead of using one-time keys. Additionally, we modeled and analyzed the proposed authentication algorithm based on VE using the security verification tool ProVerif. The security verification using ProVerif showed that there are no effective attacks. However, similar to theoretical attacks, reusing the key gives Eve the opportunity to perform critical attacks. In other words, the algorithm is essentially secure unless the encryption key is reused.

In the previous paper, it was shown that the implementation using the one-time pad is quick. By applying the proposed fast and secure VE-based algorithm to authentication in IoT devices, it is possible to maintain convenience while protecting the personal information of users. Furthermore, our algorithm is versatile because it can process a variety of personal information. Therefore, our algorithm can contribute to improving the authentication of IoT devices.

## 6. Patents

The following patent of the VE-based authentication algorithm proposed in [16] is in effect:

"ENCRYPTED DATA PROCESSING SYSTEM AND PROGRAM." JP Appl. No. 2019-560999.; US Appl. No. 16/955,739. PCT Filed: Dec. 11, 2018. Inventors: Satoshi Iriyama, Maki Kihara. Applicant: Tokyo University of Science Foundation.

**Author Contributions:** Conceptualization, S.I., M.K.; validation, M.K.; writing—original draft preparation, M.K.; writing—review and editing, M.K., Editage; All authors have read and agreed to the published version of the manuscript.

**Funding:** This research received no external funding.

**Data Availability Statement:** Not applicable.

**Conflicts of Interest:** The authors declare no conflict of interest.

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
