# Peer review of "Security Verification of an Authentication Algorithm Based on Verifiable Encryption"

_information, doi:10.3390/info14020126_

Round 1

Reviewer 1 Report

The paper presents an extended version of the authors  previous research to prove the verification process security of the developed authentication algorithm based on verifiable encryption. The authors discuss the robustness of the proposed algorithm against various theoretical attacks. The paper is well written and includes comprehensive details about the algorithm and its robustness, which are considered an advantage, and make it readable by non-specialist audiences. My main concern about the background and related work is that I believe they should be included to some extent to make the paper self-explanatory.

Best regards,

Author Response

Dear Reviewer 1,

We wish to express our appreciation to the reviewers for their insightful comments on our manuscript.
The comments have helped us significantly improve the manuscript.
Please consider the attached manuscript that has been revised according to the reviews.
A list of changes is added at the end of the manuscript.
Response to reviewer's comments follows this letter.

  •  English language and style are fine/minor spell check required
    >We thank the reviewer for the careful review. We have carefully reviewed the manuscript, and corrected typos.
  • The paper presents an extended version of the authors previous research to prove the verification process security of the developed authentication algorithm based on verifiable encryption. The authors discuss the robustness of the proposed algorithm against various theoretical attacks. The paper is well written and includes comprehensive details about the algorithm and its robustness, which are considered an advantage, and make it readable by non-specialist audiences.
    >We thank the reviewer for these excellent comments. 

  • My main concern about the background and related work is that I believe they should be included to some extent to make the paper self-explanatory.
    >We agree with you and have incorporated this suggestion throughout our manuscript. As suggested, we have added the article to the Reference.

In addition, as suggested by other reviewer, we have added the following:

  1. We added the details of definition of Hamming distance. 
  2. We compared other authentication schemes.

Again, thank you for giving us the opportunity to strengthen our manuscript with your valuable comments and queries.
We have worked hard to incorporate your feedback and hope that these revisions persuade you to accept our submission.

Sincerely,

Maki Kihara

Department of Information Science, Tokyo University of Science Yamazaki 2641, Noda, Chiba, 278-8510, Japan; +81-4-7124-1501
[email protected]

Reviewer 2 Report

My comments on the work are mostly textual:

1) Introduction can be significantly expanded. It might be worth adding a Related work section.

2) The bibliography can also be significantly extended.

Minor remarks:

1) the abstract contains literature references. Same with Conclusions.

2) there are errors in the text various editing errors, e.g., line 85: "Hamming distance" instead of "hamming distance."

Author Response

Dear Reviewer 2,

We wish to express our appreciation to the reviewers for their insightful comments on our manuscript. The comments have helped us significantly improve the manuscript. Please consider the attached manuscript that has been revised according to the reviews. A list of changes is added at the end of the manuscript. Response to reviewer's comments follows this letter.

  • Introduction can be significantly expanded. It might be worth adding a Related work section.
  • The bibliography can also be significantly extended.
    >We agree with you and have incorporated this suggestion throughout our manuscript. As suggested, we have expanded the Introduction, and added the article to the Reference.
  • The abstract contains literature references. Same with Conclusions.
  • There are errors in the text various editing errors, e.g., line 85: "Hamming distance" instead of "hamming distance.
    >We thank the reviewer for the careful review. We have carefully reviewed the manuscript, and corrected typos.

In addition, as suggested by other reviewer, we have added the following:

    1. We added the details of definition of Hamming distance. 
    2. We compared other authentication schemes.

Again, thank you for giving us the opportunity to strengthen our manuscript with your valuable comments and queries. We have worked hard to incorporate your feedback and hope that these revisions persuade you to accept our submission.

Sincerely,

Maki Kihara
Department of Information Science, Tokyo University of Science Yamazaki 2641, Noda, Chiba, 278-8510, Japan; +81-4-7124-1501
[email protected]

Reviewer 3 Report

The following problems need to be addressed to improve the quality of the paper.

(1) Normally, the one-time pad cryptosystem is impractical and seldom is used, because the one-time pad cryptosystem requires the key length with the same size of the plaintext. The reviewer does not know why this work uses the one-time pad cryptosystem for Internet of Things.

(2) In section 1, it claims “In the implementation using the one-time pad, it has been shown that the speed of authentication is as fast as 0.1 ms or less even with a text length of 8192 bits”. However, the paper does not provide the experiment for authentication operation.

(3) The definition of the hamming distance V is unclear, that is, how to compute (p1p2)i is not described.

(4) The contribution of this work is not significant, when consider “M. Kihara; S. Iriyama. New authentication algorithm based on verifiable encryption with digital identity. Cryptography, 2019, 3, 19”. The difference of them should be highlighted.

(5) The writing quality of the paper is not good. The references seldom appear in Abstract. The grammar problems and typesetting errors need to be corrected. The examples include “network technologies has also advanced significantly” in Page 1, “lightweight and secure authentication algorithms for IoT devices is required” in Page 2, and “In previous papers [1], it have been shown that implementations using this one-time pad are very fast” in Page 11.

Author Response

Dear Reviewer 3,

We wish to express our appreciation to the reviewers for their insightful comments on our manuscript. The comments have helped us significantly improve the manuscript. Please consider the attached manuscript that has been revised according to the reviews. A list of changes is added at the end of the manuscript. Response to reviewer's comments follows this letter.

  • Extensive editing of English language and style required 
    >We thank the reviewer for the careful review. We have carefully reviewed the manuscript, and corrected typos.
  • (1) Normally, the one-time pad cryptosystem is impractical and seldom is used, because the one-time pad cryptosystem requires the key length with the same size of the plaintext. The reviewer does not know why this work uses the one-time pad cryptosystem for Internet of Things.
    > We thank the reviewer for this insightful comment. We used the one-time pad because it is a cryptographic system that belongs to VE. Our algorithm does not require key distribution, which is a problem with one-time pads. In other words, the personal information used for authentication can be protected with perfect secrecy.
    We agree with the reviewer that the key length issue. The algorithm is not suitable for low-resource devices if applying very large plaintexts. However, arbitrary plaintext that is convertible into a binary sequence can be applied to the algorithm because our algorithm does not require specification the plaintext form. We consider if personal information that is not too long is chosen, the problem of key length can be ignored.
  • (2) In section 1, it claims “In the implementation using the one-time pad, it has been shown that the speed of authentication is as fast as 0.1 ms or less even with a text length of 8192 bits”. However, the paper does not provide the experiment for authentication operation.
    > We appreciate the reviewer's concerns on this point and the opportunity to clarify this point. We believe the reviewer is mistaken on this admittedly difficult point. This result is shown in previous paper “M. Kihara; S. Iriyama. New authentication algorithm based on verifiable encryption with digital identity. Cryptography, 2019, 3, 19”, and we do not measure the speed performance in this study.
  • (3) The definition of the hamming distance V is unclear, that is, how to compute (p1p2)i is not described.
    > We wish to thank the reviewer for this comment. We agree that this point requires clarification and have added the definition and some additional description in Section 2.
  • (4) The contribution of this work is not significant, when consider “M. Kihara; S. Iriyama. New authentication algorithm based on verifiable encryption with digital identity. Cryptography, 2019, 3, 19”. The difference of them should be highlighted.
    > We appreciate these helpful suggestions. The aim of the manuscript is to investigate the security of the algorithm proposed in “M. Kihara; S. Iriyama. New authentication algorithm based on verifiable encryption with digital identity. Cryptography, 2019, 3, 19” further. That is, we consider that this manuscript is a continuation version of previous study.
  • (5) The writing quality of the paper is not good. The references seldom appear in Abstract. The grammar problems and typesetting errors need to be corrected. The examples include “network technologies has also advanced significantly” in Page 1, “lightweight and secure authentication algorithms for IoT devices is required” in Page 2, and “In previous papers [1], it have been shown that implementations using this one-time pad are very fast” in Page 11.
    >We agree with you and have incorporated this suggestion throughout our manuscript. As suggested, we have carefully reviewed the manuscript, and corrected.

In addition, as suggested by other reviewer, we have expanded the Introduction and added the article to the Reference.

Again, thank you for giving us the opportunity to strengthen our manuscript with your valuable comments and queries. We have worked hard to incorporate your feedback and hope that these revisions persuade you to accept our submission.

Sincerely,

Maki Kihara
Department of Information Science, Tokyo University of Science Yamazaki 2641, Noda, Chiba, 278-8510, Japan; +81-4-7124-1501
[email protected]

Reviewer 4 Report

This manuscript proposed an authentication algorithm based on Verifiable Encryption and verified security. However, for publication, the following issues must be resolved.

1. Title: A clear title that can specify the characteristics and objectives of the study needs revision.

2. Introduction: The authors should introduce in detail the broader theories and background studies on OTP and VE-based authentication algorithms.

3. The authors must add parts that can be compared with existing studies in any way, whether qualitative or quantitative.

4. Critical analysis of the finding and limitation is missing. This would help the readers to further improve the study.

5. Add recent relevant studies to References.

Author Response

Dear Reviewer 4,
We wish to express our appreciation to the reviewers for their insightful comments on our manuscript. The comments have helped us significantly improve the manuscript. Please consider the attached manuscript that has been revised according to the reviews. A list of changes is added at the end of the manuscript. Response to comments follows this letter.

  • This manuscript proposed an authentication algorithm based on Verifiable Encryption and verified security. However, for publication, the following issues must be resolved.
    1. Title: A clear title that can specify the characteristics and objectives of the study needs revision.
    >We appreciate the reviewer's concerns on this point and the opportunity to clarify this point. We consider our original title correct because the aim of this manuscript is to verify the security of our authentication algorithm based on Verifiable Encryption proposed in [M. Kihara; S. Iriyama. Cryptography, 2019, 3, 19]. Thus, we would like to retain the original title.
  • 2. Introduction: The authors should introduce in detail the broader theories and background studies on OTP and VE-based authentication algorithms.
  • 5. Add recent relevant studies to References.
    > We agree with you and have incorporated this suggestion throughout our manuscript. As suggested, we have expanded the Introduction, and added the article to the Reference.
  • 3. The authors must add parts that can be compared with existing studies in any way, whether qualitative or quantitative.
    >We thank the reviewer for this insightful comment. As suggested, we compared other authentication schemes based on arbitrary cryptosystems in section4.3.
  • 4. Critical analysis of the finding and limitation is missing. This would help the readers to further improve the study.
    >We appreciate this helpful suggestion. We added the issues of our algorithm such as a problem of the key length of the one-time pad.

In addition, as suggested by other reviewer, we have added the details of definition of Hamming distance.

Again, thank you for giving us the opportunity to strengthen our manuscript with your valuable comments and queries. We have worked hard to incorporate your feedback and hope that these revisions persuade you to accept our submission.

Sincerely,

Maki Kihara
Department of Information Science, Tokyo University of Science Yamazaki 2641, Noda, Chiba, 278-8510, Japan; +81-4-7124-1501
[email protected]

Round 2

Reviewer 3 Report

Section 3.1 actually proposes an authentication scheme. However, in the verification operation, it is rare to employ the secure communication channel between Alice and Bob.  It needs to take more explanation. In addition, the server may not know Alice in the registration operation, because Bob only sends c1.

English language still needs to be polished. The writing mistakes include “4.1.3. Chosen-plaintext attack and Adaptive chosen-plaintext attack” in line 209 (adaptive?) and “if the same key is not reu sed” in line 388.

Author Response

Dear Reviewer 3, 

We wish to express our appreciation to the reviewers for their insightful comments on our manuscript.
The comments have helped us significantly improve the manuscript.
Please see the attachment.

A list of changes is added at the end of the manuscript (change points are in blue). Response to reviewer's comments follows this letter.

<Response to Reviewer 3 Comments>

Point 1: Section 3.1 actually proposes an authentication scheme. However, in the verification operation, it is rare to employ the secure communication channel between Alice and Bob.  It needs to take more explanation.

Response 1: We appreciate these helpful suggestions. We have changed the structure of section 3 because it is actually just introduction of the implementation of the previous paper. Specifically, we have deleted Section 3.1 and added the some sentences.

Point 2: In addition, the server may not know Alice in the registration operation, because Bob only sends c1.

Response 2: We appreciate the reviewer's concerns on this point. Just to confirm, your comment means that c1 and Alice need to be linked with something like an ID, is it correct?

If this is correct, then the following is our response:

We appreciate the reviewer's interest in additional information on database management by S. In the case of operations where there are not many users, we think that there is no particular problem even if all ciphertexts stored in S are verified. Conversely, in the case of operations where there are a large number of users, we think that the management using ID in the database is required. However, since this is a practical issue and not a theoretical one, we consider that this may not be necessary, and therefore wish to retain the original text.

Point 3: English language still needs to be polished. The writing mistakes include “4.1.3. Chosen-plaintext attack and Adaptive chosen-plaintext attack” in line 209 (adaptive?) and “if the same key is not reu sed” in line 388.

Response 3: We thank the reviewer for the careful review.  We have corrected the point you pointed out and we have carefully reviewed the manuscript, and corrected typos again.

Again, thank you for giving us the opportunity to strengthen our manuscript with your valuable comments. We have worked hard to incorporate your feedback and hope that these
revisions persuade you to accept our submission.
Sincerely,
Maki Kihara
Department of Information Science, Tokyo University of Science Yamazaki 2641, Noda, Chiba,
278-8510, Japan; +81-4-7124-1501
[email protected]
